# Diffusion of White-Hat Botnet Using Lifespan with Controllable Ripple Effect for Malware Removal in IoT Networks

**DOI:** 10.3390/s23021018

**Published:** 2023-01-16

**Authors:** Mohd Anuaruddin Bin Ahmadon, Shingo Yamaguchi

**Affiliations:** Graduate School of Sciences and Technology for Innovation, Yamaguchi University, 2-16-1 Tokiwadai, Ube 755-8611, Japan

**Keywords:** white-hat botnet, botnet defense system, malware, lifespan, ripple effect, IoT network

## Abstract

Self-propagating malware has been infecting thousands of IoT devices and causing security breaches worldwide. Mitigating and cleaning self-propagating malware is important but challenging because they propagate unpredictably. White-hat botnets have been used to combat self-propagating malware with the concept of fight fire-with-fire. However, white-hat botnets can also overpopulate and consume the resource of IoT devices. Later, lifespan was introduced as a self-destruct measure to restrain white-hat botnets’ overpopulation, but unable to change based on real-time situations. This paper proposes a method for diffusing white-hat botnets by controlling lifespan. The main contribution of this paper is that the method uses a dynamic lifespan that increases and decreases based on the congregation’s situation of the self-propagating malware and white-hat botnets. The method tackles the problem of overpopulation of white-hat botnets since they can self-propagate by controlling the ripple effect that widens the white-hat botnet’s diffusion area but suppresses the number of white-hat botnets to achieve a ’zero-botnet’ situation. The effectiveness in reducing the overpopulation rate was confirmed. The experiment result showed that the ripple effect could reduce the number of white-hat botnets in the network by around 80%, depending on different control parameters.

## 1. Introduction

Many initiatives were taken to cease the spread of malware and botnets in an infected network. Botnets such as Mirai [1], which was founded in 2016, can turn infected devices into zombies to perform large-scale Distributed Denial of Service (DDoS) attacks. It can attack IoT devices, such as cameras, that mostly run with default usernames and passwords. Later in 2018, a well-known banking trojan that was founded in 2014 called Emotet [2] became active again. It had evolved to become the door opener for other malware, such as spyware and ransomware. It is then used as a ’mercenary botnet’ for other malicious programs to exploit and download additional malware. The advancement of botnets’ threat cause malware infection to worsen by not only making devices into zombies but also into a tool for extorting money from their victims. Therefore, there is an urgency to stop the spread of malware as soon as possible.

Some researchers have proposed the method of using white-hat botnets. White-hat botnets [3] is regarded as a ‘vigilante’ program that helps to mitigate and clean up malware. In 2014, Wifatch [4] appeared as one of the first good botnets that secure the system from threats. Wifatch was used to perform updates from its peer-to-peer network to prevent malicious malware from infecting a device. It is known that Wifatch does not contain any malicious payloads. Therefore, it was used by security experts to patch security flaws such as open Telnet ports. The advantage of Wifatch is that it does not use any zero-day exploits or backdoors. It basically uses Telnet and interacts with its peers. It does not have any heavy malicious payload. However, Wifatch cannot respond to any changes, such as finding new threats in the network, unless modified by its creator before being released. Therefore, Wifatch is a static botnet. Then, in 2016, Hajime [5] appeared as another white-hat botnet. It is created to target low-level security, such as default passwords, and inform the flaws. It utilizes several attacks similar to its malicious rival Mirai. The advantage of Hajime is that it spreads quickly on many unsecured IoT devices, such as cameras, to secure the device from Mirai. However, Hajime resides permanently unless the devices are rebooted. This makes Hajime stays for a long time and overpopulates into a massive network. To solve a white-hat botnet’s overpopulation and residency problem, Yamaguchi [6] proposed lifespan to control how long white-hat botnets can exist. Lifespan is used to set the self-destruct time of white-hat botnets. Then, various observability, controllability, and basic commands of the white-hat botnets were proposed [7]. Here, white-hat botnets are equipped with a lifespan, so they will not permanently reside in the network. However, the problem persists where white-hat botnet overpopulates because it is hard to observe how many white-hat botnets self-propagate during their existence.

An IoT network may contain computers, sensors, and actuators. All devices transmit data through a router, access point, or mobile ad-hoc network. Regardless of the network topology, broadcast protocol, and devices, any malware may exploit a network if at least one device is vulnerable. Therefore, a three-technique method which is detection [8], prevention [9], and mitigation [10], is required. The detection method focuses on detecting malware through several methods, such as signature-based or machine-learning methods. The prevention method focuses on preventive measures, whereas the method focuses on early defensive measures to block malware from invading a network. Next, the mitigation method focuses on reducing the threat of malware that has already infected a system. However, although malware can be removed from individual devices, ensuring all network devices are clean is hard. This leads to the fourth technique, called clean-up. In general, the clean-up technique is regarded as a part of mitigation where malware is removed from a system. This is because, although malware was removed from a network, white-hat botnets may reside in the network for a long time. Botnets consume each device’s resources and may expose an open port for another attack. In mitigating malware attacks, we regard the clean-up technique as a proper technique of cleaning both malware and white-hat botnet.

In this paper, we focus on the clean-up technique. We diffuse a white-hat botnet into an infected network to exterminate malware. After the malware extermination using a white-hat botnet, our method ensures that all connected nodes are clean from malware and the white-hat botnet itself. The main objective is to achieve a ’zero-botnet’ situation where the network is free from malware or white-hat botnet. The major contributions of this paper are:(i)The method proposes a control method of lifespan for white-hat botnets. The lifespan changes according to the state of the white-hat nodes, where the lifespan can increase or decrease.(ii)The method controls the ripple effect when diffusing white-hat botnets. The characteristics of the ripple effect are that the range and concentration of white-hat botnets change depending on the parameter settings.(iii)The evaluation of dynamic lifespan to the diffusion strategies using a white-hat botnet.

Each white-hat botnet has a lifespan that changes based on the density of the white-hat botnet network. After the extermination of malware at the leaf nodes, the white-hate botnet will self-destruct at the end of its lifespan. In the evaluation, we show that by using a controlled lifespan, a white-hat botnet can exterminate all malware in the network and disappear faster to achieve a ’zero-botnet’ situation compared to a fixed lifespan.

## 2. Related Works

We can distinguish the technique of handling malware attacks into four techniques known as detection, prevention, mitigation, and clean-up. Here, we briefly summarize some representative-related works that lead to the clean-up technique where we position our research. Many types of research did studies on the first three techniques, but they gave very less focus to the fourth approach, which is the clean-up approach. Table 1 shows the related works grouped by categories that distinguish our method using BDS (Botnet Defense System) launcher. There are four main techniques for defending IoT networks from malware: detection, prevention, mitigation, and clean-up. Some techniques may depend on each other, and others may be taken independently.

The first technique is detection. This technique involves detecting botnets or malware by identifying their signatures and evaluating the applications’ hash to validate the legitimacy of any update or modifications. Namanya et al. proposed a hash scoring technique of portable executable files [11]. The technique combines four types of file similarity hashing that have improved accuracy compared to individual hashing. Kakisim et al. [12] proposed a Metamorphic malware identification using engine-specific patterns. Rehmann et al. [13] proposed a machine learning-assisted signature and heuristic-based detection of malware. Vasan et al. [14] and Liu et al. [15] proposed machine-learning for classifying malware. Botacin et al. [16] proposed a hardware-assisted signature-based malware detection. Their method is memory efficient for mobile devices, which only takes a small amount of storage and reduces the CPU’s workload. Other than signature-based detection using hash, machine learning has been widely applied. Yadav et al. and Hussain et al. [17] used a machine-learning method to detect malware. Their techniques show promising results in detecting malware compared to the traditional method, which is less effective against polymorphism and code obfuscation. These advanced detection techniques enable the second technique, which is the prevention technique. Other signature-based works are given in Ref. [28] and by Witchmann et al. [29]. Other works related to machine learning are given by Martinelli et al. [24], Kouliaridis et al. [30], Moussas et al. [31], and Yadav et al. [32].

The second technique is prevention. Prevention includes moving target defense, DNS list analysis, and pre-deployment simulation. Albanese et al. [18] and Amich et al. [19] proposed a technique called moving target defense (MTD) to prevent botnet attacks. MTD continuously changes or shifts a system’s attack surface to complicate the attackers. Their method can limit the ability of the botnet to exploit the target system. Nadler et al. [28] focuses on the DNS anti-malware list (DNSAML) of DNS services used by anti-malware solutions. They demonstrated attacks that may cause anti-malware agents to ignore threats and proposed a set of countermeasures. The prevention technique is quite a difficult technique. Therefore, most systems are more likely to be exposed to botnets and malware. Hwang et al. [20] proposed a simulator for evaluating preventive measures before deploying the measures in the real world. The tool can render massive IoT networks so that the effectiveness of malware detection and prevention can be evaluated. Some other preventive measures were also proposed by Sajjad et al. [21], Dinakarro et al. [33], and Ajmal et al. [22]. However, preventive techniques are only one of the defense lines leading to the mitigation technique.

The third technique is the mitigation technique. The mitigation technique is taken when a network is exposed to malware attacks. This technique requires proper tactics and strategies to ease the impact of attacks by botnets and malware. Yamaguchi [6] introduced the concept of fighting fire with fire. The method uses a white-hat botnet to fight malware. The white-hat botnet is a type of IoT worm called Hajime that cleans up vulnerabilities in IoT devices. The fight between white-hat botnets and Mirai can be represented with an agent-oriented Petri net model called PN^2^. Yamaguchi [26] also introduced control strategies and basic commands for cleaning-up malware with white-hat botnets. However, once a botnet node occupies a host, it also occupies the resources and becomes a burden to the host. Therefore, white-hat botnets should not reside long in the network. Here, lifespan shows a promising method to allow white-hat botnets to disappear after a certain period. Later, Yamaguchi et al. [34] proposed a botnet defense system that utilizes machine learning. Machine learning is used along with the divide and conquers algorithm to diffuse the white-hat botnet into the system. Another mitigation method was proposed in Than Vu et al. [35] and Mahboubi et al. [36]. Chu et al. [23] proposed clustering classification mining and an algorithm to countermeasure botnets using machine learning.

The fourth technique is clean-up. Despite the effort to detect, prevent and mitigate botnet attacks, once botnets infect a network, they can linger and persist for a long time. This is due to the existence of botmasters and command and control servers that dynamically change their IP addresses even though their domain is known. Moreover, botnets infect any vulnerable targets found within their scanning range. In 2018, the Council on Foreign Relations [37] brought up the idea of zero-tolerance policies for botnets. They started the idea of cleaning up the internet from botnets. Then, in 2022, the idea is supported by Kepner et al. [25]. They stated that botnets are usually cleansed long after the initial infection. The process of botnet’s growth consists of plan, stage, infection, spread, detection, and cleanse. Once malware or botnet are detected, widespread clean-up can take place. However, this process repeats indefinitely. They proposed the idea of *observe-pursue-counter approach*. They gave the key parameter of the defense system as the detection time from initial infection to detection $t_{detect}$. The smaller the $t_{detect}$, the more effective the system will be. If $t_{detect}$ can be reduced significantly, the impact of botnet spread can be reduced. Therefore, the most effective way is to “defend forward”. However, there are still very few works related to the forward defense mechanism, such as by Wichman et al. [29]. They used infection markers marked by malware to generate vaccines on the infected computer without input from human experts.

Table 1 shows our method proposed by Yamaguchi et al. [26] is used in the fourth stage, which is the clean-up stage. The method uses white-hat botnets to exterminate malware in a network. Then Pan et al. [27] proposed zoning tactics to exterminate malware using clustering. The white-hat botnet is diffused into the network using a launcher and placed in a specific position to form a zone for clean-up or mitigate the attack. Both methods by Yamaguchi et al. and Pan et al. proposed the idea of using lifespan in their strategies. The idea of using lifespan has increased the flexibility of controlling tactics when diffusing white-hat botnets. Bin Ahmadon et al. [38] showed that more effective tactics could be proposed by changing the length of lifespan. Therefore, our work proposed a method that utilizes a dynamic lifespan. We showed that by allowing the lifespan of the white-hat botnet to change dynamically, we could achieve a better result during clean-up. Doroudi et al. [39] proposed a clean-up strategy for servers. Markov chain was used to evaluate malware cleanup policies. They improve clean-up policies by queue length information. Chakravarty et al. [40] stated that most of the works in malware mitigation focus on the defensive measure, which is to use a firewall, scanning techniques, and security compartmentalization. That is to say most work focus on prevention and detection.

## 3. Approach of Diffusion Based on Dynamic Lifespan

### 3.1. Research Problem

In our perspective, we diffuse white-hat botnets that can detect and remove malware in a network and enable the white-hat botnet to exist for a certain period. The diffusion is controlled by lifespan that changes based on the density of white-hat botnets and malware. The forward defense strategy will be used to exterminate all malware and to withdraw all white-hat botnet nodes, which we call a “zero botnet” state.

Specific scenarios and network topologies may affect the effectiveness of mitigating malware attacks. In the research problem, we need to determine the best strategy for mitigation and clean-up. Eventually, the strategy should lead to successful diffusion called a ‘zero-botnet’. We believe that there is a trade-off between the population of white-hat botnets and lifespan. It is important to avoid the overpopulation of white-hat botnets after a successful clean-up in a network.

We will determine the most effective lifespan control pattern in specific scenarios where nodes are dense and scattered. Lifespan will dynamically change, but it is important to exterminate malware and leave no white-hat botnet in the network while suppressing the spread of malware in a shorter time. Here, to control the white-hat botnet to clean up infected nodes, we utilize a ratio called lifespan ratio. The lifespan ratio can be changed before launching the white-hat botnet into the network.

**Definition** **1**(Lifespan Ratio). *Lifespan ratio r is the multiplication value for increasing and decreasing the current lifespan L. Ratio r can be denoted as α for increase ratio and β for decrease ratio.*

The problem can be defined as a problem to decide the lifespan ratio when diffusing white-hat botnets. Based on the following problem, we denote the increase ratio of *r* is denoted by α and the decrease ratio is denoted by β. The output of the problem can be defined as a clean network where there are no infected nodes.

**Definition** **2**(White-Hat Botnet Diffusion Problem).**Input:** *Network nodes $N\;=\;\{n_{1},n_{2},\cdots ,n_{k}\}$ (∃n∈N:n∈M∪W, M is a set of infected nodes and W is a set of white-hat botnets with initial lifespan L), increase ratio α, decrease β***Output:** *Network with nodes $N^{\prime }\;=\;\left\{n_{1},n_{2},n_{3},\cdots ,n_{k}\right\}$ where ∀n∈N:n∉M∪W*


The given problem takes input from nodes $N\;=\;\{n_{1},n_{2},n_{3},\cdots ,n_{k}\}$ in a network. Initially, there exists a set of infected nodes M and white-hat botnets W deployed in the network. The objective of this paper is to achieve a ’zero-botnet’ situation where eventually no malware of white-hat botnets exists. Therefore, the problem is to clean the network from white-hat botnets where M=ϕ and W=ϕ.

### 3.2. Lifespan-Based Control Method with Ripple Effect

To control lifespan, we propose the increase ratio α and β. They decide how long the increase and decrease of the lifespan should be when a certain condition is satisfied.

Initial Lifespan *L*: Initial lifespan *L* is the initial lifespan given to a white-hat botnet when it spreads to another node.Increase ratio α: Increase ratio is the rate of increase from initial lifespan *L* such that $L^{\prime }\;=\;L+\alpha$.Decrease ratio β: Decrease ratio is the rate of decrease from initial lifespan *L* such that $L^{\prime }\;=\;L-\beta$.

Depending on the infection situation, the effectiveness of cleaning up the network *N* from malware depends on the initial lifespan *L*, increase ratio α, and decrease ratio β value. Here, if we set the initial lifespan along with α and β values. Therefore, we focus on manipulating the increase and decrease ratio *r* when deploying white-hat botnets.

The overview of the control method is shown in Figure 1. In a network *N* where each node $n_{k}$ is a white-hat botnet, normal node, or an infected node, the objective is to clean up the network from the infected node. In Figure 1, normal nodes are shown as white circles, white-hat botnet nodes as blue circles, and infected nodes as orange circles. Any transmission signals a white-hat botnet node (blue circle) to identify connected nodes and verify their status. The green arrows represent the signals to increase the lifespan of any white-hat botnet nodes which are connected to any infected botnet node. The red arrows represent the signal of decreasing the lifespan when the white-hat botnet receives a transmission from any white-hat botnet node.

In a network environment, network nodes can exist in a dense and sparse network. Some of the networks are also constructed in a mesh network. In a dense mesh network, infection by malware may occur more frequently and faster. The effect of infection in a dense network is larger than in a sparse network. This is because of the density of the nodes. Figure 1 shows an example of a dense mesh network. Once an infected node infected at least one node, the spreading occurs swiftly across the whole network. In the previous research [38], in order to counter the infection, we utilized a white-hat botnet with lifespan and diffuse the white-hat botnet nodes across the network.

Here, we introduce the idea of a dynamic lifespan. Dynamic lifespan can increase or decrease according to the environment. The advantage of using dynamic lifespan is that once all infected nodes have been cleaned up, all white-hat botnet nodes disappear once their life reaches its lifespan. However, there are cases where the diffusion is successful or unsuccessful. This is due to the changes in the number of white-hat botnet nodes and infected nodes because white-hat botnets reduce over time when it reaches their own lifespan. To diffuse white-hat botnets successfully we need a strategy to control lifespan in white-hat botnets.

This paper proposes a novel mechanism that leverages the congregation density between nodes called ripple effects. The ripple effect in a network is regarded as the continuation and spreading of an infection. This method leverages the ripple effect for spreading white-hat botnet nodes to mitigate and clean up infected nodes. Lifespan can be used to control the ripple effect.

The ripple effect can be controlled with ratio α and β by considering the status of the nodes in the network i.e., infected or clean. Figure 2 shows the mechanism of dynamic lifespan. Figure 2a shows an illustration of lifespan over time. Lifespan does not change when there is no connection between infected nodes or white-hat botnets. Figure 2b lifespan will increase by ratio α when a white-hat botnet node detected an infected node. Figure 2c illustrates lifespan will decrease by ratio β when a white-hat botnet node is connected with another white-hat botnet node. Figure 3 illustrates the mechanism of remaining life. Figure 3a shows remaining life decreases over time starting from the current or initial lifespan. Figure 3b illustrates the increase of remaining life by α if a white-hat botnet node detected an infected node. Figure 3c illustrates remaining life decreases by β when a white-hat botnet node congregates with another white-hat botnet node.

The illustration of our approach is shown in Figure 4. The lifespan $l_{n}$ is shown by the blue color of each node. The darker blue represents a longer lifespan. $l_{n}$ is controlled by initial lifespan *L*, increase ratio α and decrease ratio β.

First, we introduce the following parameters. These parameters take into account the environment and the lifespan.

(i)Congregation density *D*: The density of white-hat botnet nodes or infected nodes that gather within the communication range.(ii)Lifespan ratio *r*: Multiplication ratio for increasing or decreasing the current lifespan.

To decide the congregation density, we use the density calculation based on the number of white-hat botnet nodes or infected nodes and the total number of connected nodes. We give the following definition:

**Definition** **3.**
*A white-hat botnet network is said to be perfectly dense if and only if the ratio of total connected nodes k and the number of white-hat botnet nodes or infected nodes m is 1.*


We can calculate the congregation density of the white-hat botnets or infected nodes. The number of adjacent white-hat botnet congregation density *D* can be calculated as in Equation (Equation 1). In Equation (Equation 1), *k* is the number of connected nodes and *m* is the number of connected white-hat nodes or infected nodes. The congregation density *D* influences the strength of the white-hat botnets i.e., the length of its lifespan. From Equation (Equation 1), we can define the congregation density of white-hat botnets as $D_{W}$ if *m* represents the number of white-hat botnet nodes and congregation density of infected nodes as $D_{M}$ if *m* represents the number of infected nodes.
(1)$$D=\frac{2m}{k(k-1)}$$

The lifespan can be adjusted to increase or decrease in length by using the control parameter *r* (in some cases α or β). For example, given the congregation density of white-hat botnet D, we can calculate the lifespan as in Equation (Equation 2). For a perfectly dense white-hat botnet network, we set the strength of the ratio to 1; otherwise, we set the strength of the ratio to the level of the congregation density D. Here, we denote *r* to represent the reduction ratio of lifespan.
(2)$$l_{n}=\left\{\begin{array}{ccc} l_{n-1}-r & for & D=1\\ l_{n-1}-\left(D\times r\right) & for & D<1 \end{array}\right.$$

We can control the botnet’s lifespan based on Equation (Equation 2). As an example, we give three cases:**White-hat botnets in Network of Botnets**: We decrease the botnet’s lifespan by 2 times i.e., r=−2, and create a new child.We can modify Equation (Equation 2) as follows:
(3)$$l_{n}=\left\{\begin{array}{ccc} l_{n-1}-2 & for & D=1\\ l_{n-1}-\left(2\times D\right) & for & D<1 \end{array}\right.$$**White-hat botnets in Network of Botnets and Malware**: We increase the botnet’s lifespan by 2 i.e., r=2 and create a new child.We can modify Equation (Equation 2) as follows:
(4)$$l_{n}=\left\{\begin{array}{ccc} l_{n-1}+2 & for & D=1\\ l_{n-1}+\left(2\times D\right) & for & D<1 \end{array}\right.$$**White-hat botnets in Network of Normal Nodes**: We normally reduce the lifespan by 1 i.e., r=1 and create a new child.We can modify Equation (Equation 2) as follows:
(5)$$l_{n}=l_{n-1}-1$$

From the derivation of Equation (Equation 2) to Equation (Equation 5), the lifespan can be controlled with ratio *r*. Since ratio *r* can increase or decrease the lifespan, *r* can be denoted as α as the increase ratio and β as the decrease ratio. We give the white-hat botnet diffusion procedure based on controlled lifespan using Algorithm 1. In Algorithm 1, we can define ratio α and β as follows:**Increase Ratio α:** Increase ratio is a ratio in which the lifespan ratio is increased when connected nodes of a white-hat botnet are infected nodes. Increased ratio strengthens the defense by building a wall of botnets with longer lifespans when encountering infected nodes.**Decrease Ratio β:** Decrease ratio is a ratio that decreases the lifespan when the connected nodes of white-hat botnets are white-hat botnets. It is used to control the congregation density to suppress the overpopulation of white-hat botnets.

Both α and β play an important role in changing the value of lifespan and remaining life. One of the characteristics of this method is the controllable ripple effect when diffusing white-hat botnets. Figure 5 illustrates the ripple effect on a surface graph with 400 nodes using Algorithm 1. The height of the surface graph shows the remaining age of every white-hat botnet node starting from the initial lifespan L=10. The range of the remaining lifespan shown in the figure is between 0 to 30 s. Figure 5a shows the initial placement at 1 s. Then, Figure 5b shows the rise of the ripple wave at 20 s. Figure 5c shows the expansion of the ripple wave after 40 s. Note that, the middle of the wave shows a lower surface because the remaining life of the white-hat botnet nodes is lower.

The effectiveness of reducing the number of infected nodes can be controlled with the size of ratio α and β. The ratios control how wide and how long the ripple effect lasts. Here, we hypothesize that if the ratio r=α=β is fixed, the effectiveness depends strongly on the initial lifespan *L*. If α increases, the number of white-hat botnets can increase significantly. However, if α increases too much, the number of white-hat botnets become too many and resides for a long time in the network. Here, we can balance the population of white-hat botnets using the decrease ratio β. Similar to α, it is important to decide the value of β.
**Algorithm 1** Diffusion of White-Hat Botnet**Input**: Network nodes $N\;=\;\{n_{1},n_{2},n_{3},\cdots ,n_{k}\}$ and W is a set of white-hat botnets, Increase Ratio α, Reduction Ratio β, congregation density $D_{W}$ and $D_{M}$
**Output:** Lifespan *L* for each node n∈N
  ▹ *Diffuse white-hat botnet*
1:**for** each node *n* in *N* **do**2:   Obtain current state S←STATE3:   Obtain neighbour state R←NSTATE4:   Set initial lifespan $L\leftarrow l_{n}$5:   Set the lifespan ratio α←α, β←β6:   **if** STATE is BOTNET and NSTATE is NORMAL **then**7:     Set NSTATE as white-hat botnet NSTATE←BOTNET8:   **end if**9:   **if** STATE is BOTNET and NSTATE is BOTNET **then**10:     Decrease the lifespan $L\leftarrow L-(D_{W}\times \alpha )$11:   **end if**12:   **if** STATE is BOTNET and NSTATE is INFECTED **then**13:     Set NSTATE as white-hat botnet NSTATE←BOTNET14:     Increase the lifespan $L\leftarrow L+(D_{M}\times \beta )$15:   **end if**  ▹ *Output lifespan of node n*16:   Output lifespan *L* for node *n*.17:**end for**


## 4. Evaluation

### 4.1. Experiment Design

We conducted a simulation of wireless sensor networks (WSN) to evaluate our method. Basically, we compare dynamic lifespan with fixed lifespan and dynamic lifespan with different parameter settings. The simulation is illustrated in Figure 6. Contiki Cooja Simulator was used to simulate the WSN environment. The simulation setting was using Sky Mote with UDGM (Unit Disk Graph Medium) transmission model. In Figure 6, the range of reception is shown as a green circle. Connected nodes are set within the transmission range (green circle). During simulation, the status is shown as three colors: blue (white-hat botnet), red (infected node), and green (normal node). Each malware and white-hat botnet will change status once malware infects another node or a white-hat botnet clears malware from the infected node.

To evaluate different lifespan-changing mechanisms, we evaluated the method with the following lifespan ratio.

(i)Fix Ratio: Fixed lifespan with a variation of 5, 10, and 20 where r=1.(ii)Adjust Increase Ratio: Increase the initial lifespan *L* within the range of 20, 15, 10, and 5 with the increase ratio α larger than ratio β where α>β such that α∈{10,9,8,⋯,1} and β∈{1,2,3,⋯,10}.(iii)Adjust Decrease Ratio: Decrease the initial lifespan *L* of 20, 15, 10, and 5 with decrease ratio α smaller than ratio β where β>α such that β∈{10,9,8,⋯,1} and α∈{1,2,3,⋯,10}.(iv)Adjust Increase-Decrease Ratio: Increase initial lifespan *L* with a variation of 5, 10, and 20 with increase ratio α and decrease ratio β such that α∪β∈{10,9,8,⋯,1}.

### 4.2. Evaluation Results

We conducted an experiment to confirm the effect of controlling lifespan to exterminate malware in IoT networks. The simulation is shown in Figure 6. The simulation tool used is Contiki Cooja 3.0 running on Linux Ubuntu with 16 GB of RAM. The simulation can cover at most 400 nodes of wireless sensor nodes. The experiment was simulated with UGDM (Unit Graph Disc Model) distance loss.

We evaluated three important parameters; increase ratio α, decrease ratio β, and initial lifespan *L*. Six types of parameters setting were considered in the evaluation.

Fixed Lifespan: The initial lifespan *L* is fixed to L=5,10,20. There is no change in the increase or decrease ratio.Low Balanced Ratio: The increase ratio and decrease ratio are set to the same value i.e., α=β=2. The value is a relatively small value.Small Attack—Small Withdrawal Ratio: The increase ratio is set to α=2 and the decrease ratio is set to β=1. The decrease ratio is slightly lower than the increase ratio.Large Attack—Small Withdrawal Ratio: The increase ratio is set to a large value α=5 and the decrease ratio is set to a small value β=1.Large Attack—Medium Withdrawal Ratio: The increase ratio is set to a large value and the decrease ratio is set to a relatively medium value r=5,α=β.Large Attack—Large Withdrawal: The increase ratio and the decrease ratio is set to the same large amount r=5,α=β.

Figure 7 shows the result of diffusion by changing the initial lifespan. The lifespan was fixed at 5, 10, and 20. The difference between the solid line and the dashed line can recognize the success of clean-up. The solid line represents the number of white hat botnet nodes. The dashed line represents the number of infected nodes. Obviously, to identify successful outcomes, we need to confirm if the solid lines went above the dashed lines or not after the number of infected nodes becomes zero. In Figure 7a, the lifespan was set at the most minimal value, which is at L=5 and a single decrease ratio α={10,5,2} was used. Based on the result, all α parameter settings were unsuccessful because the number of infected nodes in every setting exceeds the number of white-hat botnet nodes. Next, we increased the initial lifespan to L=10. The result is shown in Figure 7b The ratio of α at 10 and 5 was too large because the number of infected nodes exceeds the white-hat botnet nodes after 20 s and 25 s. Only α=2 was successful. Finally, we increased the initial lifespan to L=20. The result is shown in Figure 7c. The results showed that the decrease ratio α=10 was also still too large for the parameter settings. From the result, we can conclude that even though we increased the initial lifespan, we cannot guarantee the success of the diffusion. The diffusion should be controlled with both increase and decrease ratios α and β.

We applied α and β to the diffusion in the next experiment settings. The result is shown in Figure 8. We found that for L=5 as shown in Figure 8a, the settings of either α=5,β=1 or α=2,β=1 was not successful. At L=10, only α=5 and β=1 was successful (see Figure 8b). At L=20, either setting was successful. However, since we increased the initial lifespan to 20, the number of white-hat botnets increased and took more time to disappear. Therefore, the best parameter settings with the most minimal number of white-hat botnets and optimal clean-up were L=10,α=5,β=1.

Table 2 shows the result for important parameter settings. Setting with ’Yes’ represents successful cleanup. The result shows that the initial lifespan does not always have an advantage in the diffusion method. For initial lifespan at L=20 all experiment was successful except for *Balance Large Attack-Withdrawal* setting. For L=10, *Small Attack-Small Withdrawal* and *Large Attack-Small Withdrawal* was successful. This shows that minimizing the decrease ratio can ensure successful diffusion.

Figure 9 shows the comparison of different parameter settings, including initial lifespan *L*, increase ratio α, decrease ratio β, and the position of the white-hat botnet launcher. The lifespan that changes through time shows that the spread of white-hat botnets can be controlled with the three parameters and by deciding the position of the launcher. The simulation using 400 WSN nodes shows different outcomes. Figure 10 compares three successful parameter settings. At around 80 s or later, the ‘zero-botnet’ situation was confirmed. Parameter setting with L=10,α=5,β=1 shows the best clean-up with minimal white-hat botnets. Here, we can conclude that α>β always leads to better results compared to α=β.

The result shows that if α=β, then the white-hat botnet will remove each other before successfully removing all infected nodes. In the case of α>β, the white-hat botnet can remove infected nodes and perform self-removal more efficiently. From the evaluation, white-hat botnets acquired higher endurance before disappearing when increasing their lifespan. Based on the result, the proposed method showed improvement in minimizing lifespan with 44% improvement in reduction of the white-hat botnet and around 70% successful reduction of infected nodes in a shorter time.

### 4.3. Discussions

Payload always exists when using botnets. That is the reason we proposed the method so that botnets do not overpopulate and congregate to the whole IoT network. By controlling the lifespan and number of white-hat botnets, the method can avoid the exhaustive usage of devices’ resources.

Previous methods in utilizing BDS launchers for mitigating malware attacks focus on the idea of self-destruction using lifespan. In this idea, once all malware has been cleaned-up the white-hat botnet nodes will also disappear. The idea of self-destruction by Yamaguchi [26] is regarded as limiting the life of a white-hat botnet. The idea of lifespan was proposed. Pan et al. [27] proposed a zoning method to mitigate malware attacks, and lifespan is used to release a zone from a botnet once the lifespan has been reached. The method is effective when forming a temporary zone when needed and releasing the zone once it has been recovered.

Bin Ahmadon et al. [38] studied the method to control lifespan for controlling the diffusion of white-hat botnets. Until now, previous research has regarded white-hat botnet diffusion as *release-and-forget* strategy. However, it has a drawback where the white-hat botnet is not responsive enough when cleaning up the network. This is because the concept of *observe-pursue-counter* approach proposed by Kepner et al. [25] was not taken into consideration. To achieve ’zero-botnet’ we need responsive white-hat botnets that can become stronger when they need to attack and withdraw once they recovered a territory. Therefore, our method of using dynamic lifespan also can be applied to *observe-pursue-counter* approach.

In this experiment, we can confirm that we can also reduce the number of white-hat botnets by controlling the lifespan and the position and direction of ripple waves. Note that in this experiment, to accelerate the reduction of white-hat botnets, the lifespan ratio increases by α when neighboring nodes are white-hat botnets or infected nodes. The lifespan ratio is reduced by β when white-hat botnet nodes congregate.

Based on the result, if the network range is unknown, i.e., wireless network, the combination of increase ratio α and decrease ratio β is recommended. However, due to the limitation of the network, it is better to keep *L* at the most minimum value where all white-hat botnets can reach all infected nodes. Parameters with α>β should be optimized with the positioning of the launcher, and balanced with the initial lifespan. It is always recommended to perform a simulation using the three parameters to achieve the optimum clean-up result.

Finally, the ripple effect helps white-hat botnets to widen the cover area and clean up the network. The ripple effect also reduces the number of white-hat botnets to prevent the ‘overpopulation’ of white-hat botnet nodes in the systems. Therefore, the proposed method is suitable for achieving the ’zero-botnet’ state of particular IoT networks.

## 5. Conclusions

White-hat botnet is a powerful tool for tackling the problem of cleaning up malware in IoT systems. White-hat botnets such as Wifatch and Hajime can patch security flaws and remove malware through rebooting. However, they may reside on the system permanently and overpopulate if no proper control method is considered. The white-hat botnets themselves are static unless the creator changes their hardcoded functions.

Most white-hat botnets are based on the release-and-forget approach. Therefore, the problem of overpopulation and overtaking system resources by their payload is impractical. Here, lifespan has been identified as one of the solutions. However, botnets with static lifespans could not respond to the situation. There is a need for a dynamic lifespan that can change based on the congregation situation of white-hat botnet.

In this paper, we proposed a method that utilizes dynamic lifespans. The advantage of using a dynamic lifespan is a controllable ripple effect that effectively increases and decreases the population of white-hat botnets. At the same time, it can effectively remove malware that eventually achieves a ’zero-botnet’ situation. We proposed 3 important parameters in the proposed method to control the remaining lifespan $l_{n}$: initial lifespan *L*, increase ratio α, and decrease ratio β. The proposed method showed that the proper balance ratio of the 3 parameters could effectively allow a white-hat botnet to respond effectively in removing malware.

In future work, we will consider more advanced techniques to utilize dynamic lifespan in white-hat botnets. For example, to control two ripple effects of white-hat botnets that cancel each other so that the diffusion at multiple launcher positions can achieve optimal effect in malware removal and suppressing overpopulation. We expect dynamic lifespan usage in white-hat botnets to contribute to the strategy that has more control and feasibility for tackling self-propagating malware attacks in various network topologies. For example, to create different variants of white-hat botnets that cooperate and compete with each other to clean up the network. We will also consider adversarial attacks where we will consider and discuss adversarial attack detection. In adversarial attack, we will consider utilizing our technique to tackle the problem by sparsely positioned malicious bots (saturation attack), continuous malicious bot launching, fake malicious bots, or a combination of all attacks.

## Figures and Tables

**Figure 1 sensors-23-01018-f001:**
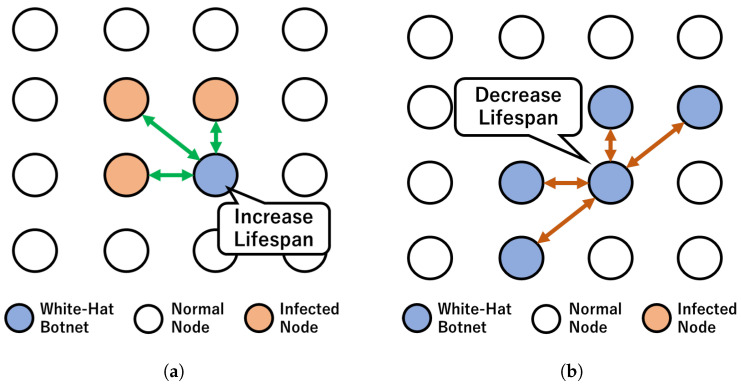
Overview of the dynamic lifespan control. (**a**) Increase lifespan when a white-hat botnet node encounters infected nodes. (**b**) Decrease lifespan when white-hat botnet nodes congregate.

**Figure 2 sensors-23-01018-f002:**
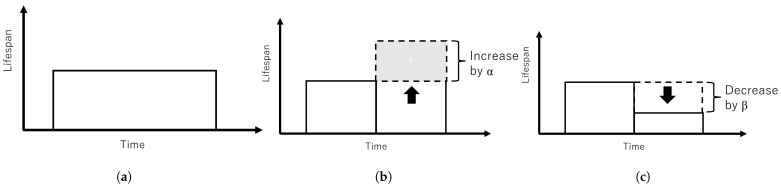
Mechanism of dynamic lifespan that changes according to node status. (**a**) Normal lifespan change. (**b**) Lifespan increases when a white-hat botnet node detected a malware node. (**c**) Lifespan decreases when a white-hat botnet node congregates with another white-hat botnet node.

**Figure 3 sensors-23-01018-f003:**
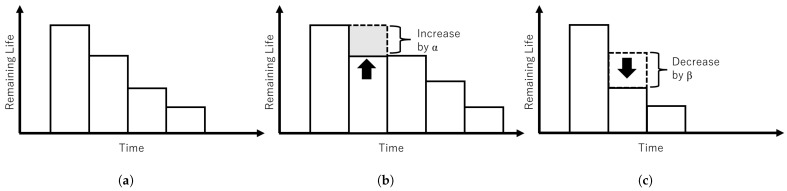
Mechanism of remaining life according to node status. (**a**) Normal white-hat botnets’ remaining life over time. (**b**) Remaining life when a white-hat botnet node detected an infected node. (**c**) Remaining life decreases when a white-hat botnet node congregates with another white-hat botnet node.

**Figure 4 sensors-23-01018-f004:**
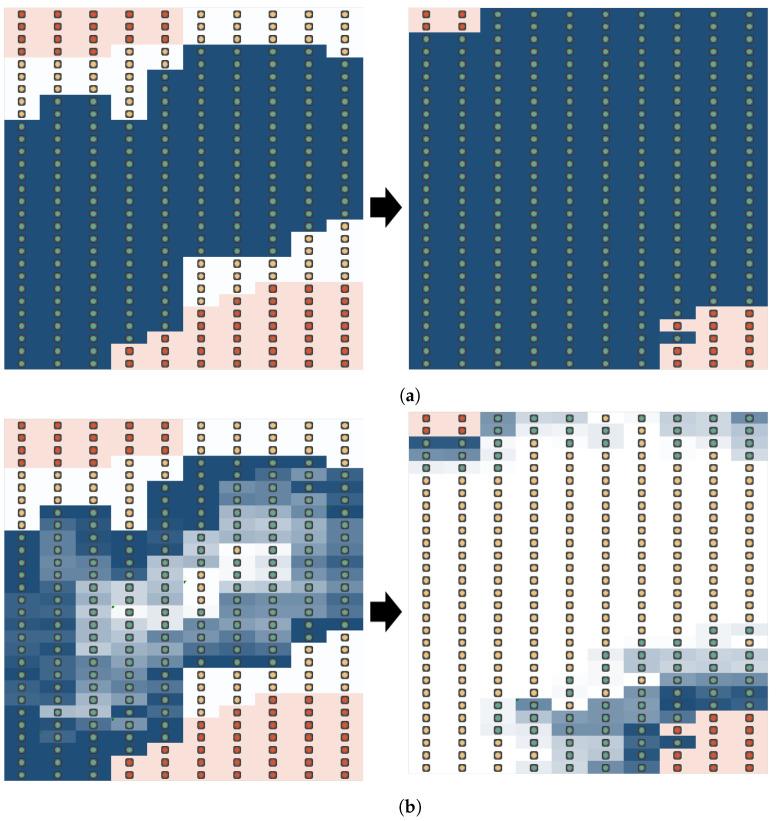
Illustration of our approach. (**a**) Illustration of conventional white-hat botnet diffusion. (**b**) Illustration of diffusion using our approach.

**Figure 5 sensors-23-01018-f005:**
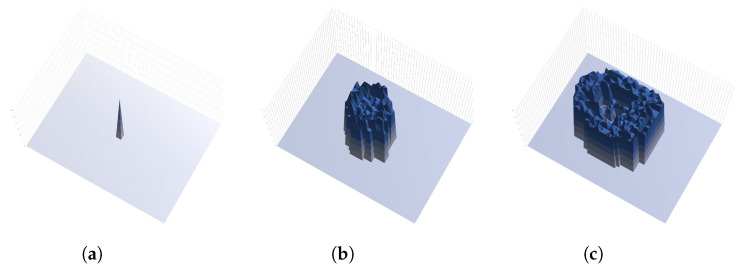
Image of the ripple effect. The height of the surface graph shows the remaining age. (**a**) Initial placement of white-hat botnet (t=1). (**b**) The rise of ripple effect (t=20). (**c**) The expansion of ripple effect (t=40).

**Figure 6 sensors-23-01018-f006:**
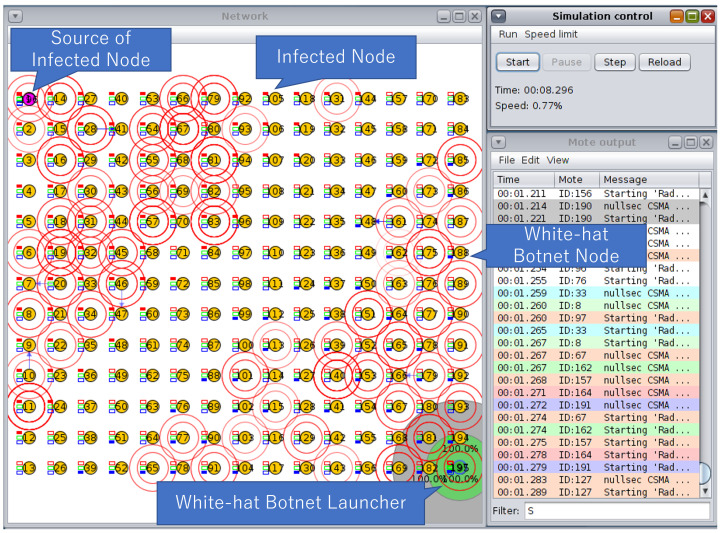
An illustration of a simulation scenario.

**Figure 7 sensors-23-01018-f007:**
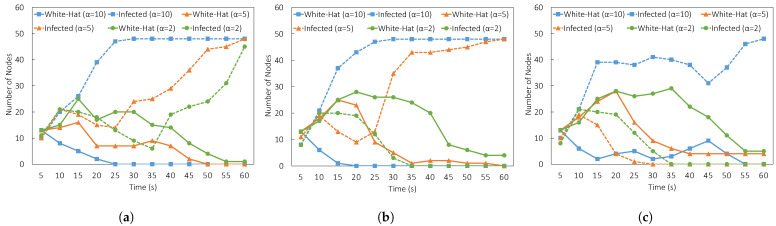
Comparison of result based on initial lifespan *L*. (**a**) Experiment result (L=5,α={10,5}). (**b**) Experiment result (L=10,α={10,5}). (**c**) Experiment result (L=20,α={10,5}).

**Figure 8 sensors-23-01018-f008:**
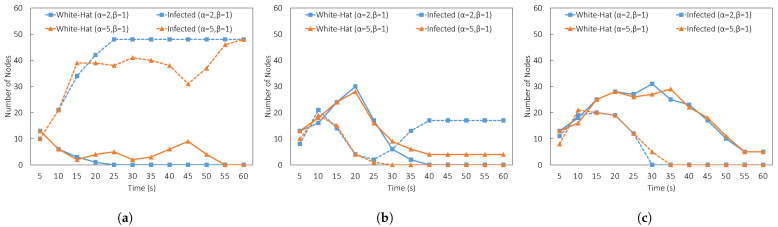
Comparison of result by changing ratio α and β where α>β. (**a**) Experiment result (L=5,α={2,5},β=1). (**b**) Experiment result (L=10,α={2,5},β=1). (**c**) Experiment result (L=20,α={2,5},β=1).

**Figure 9 sensors-23-01018-f009:**
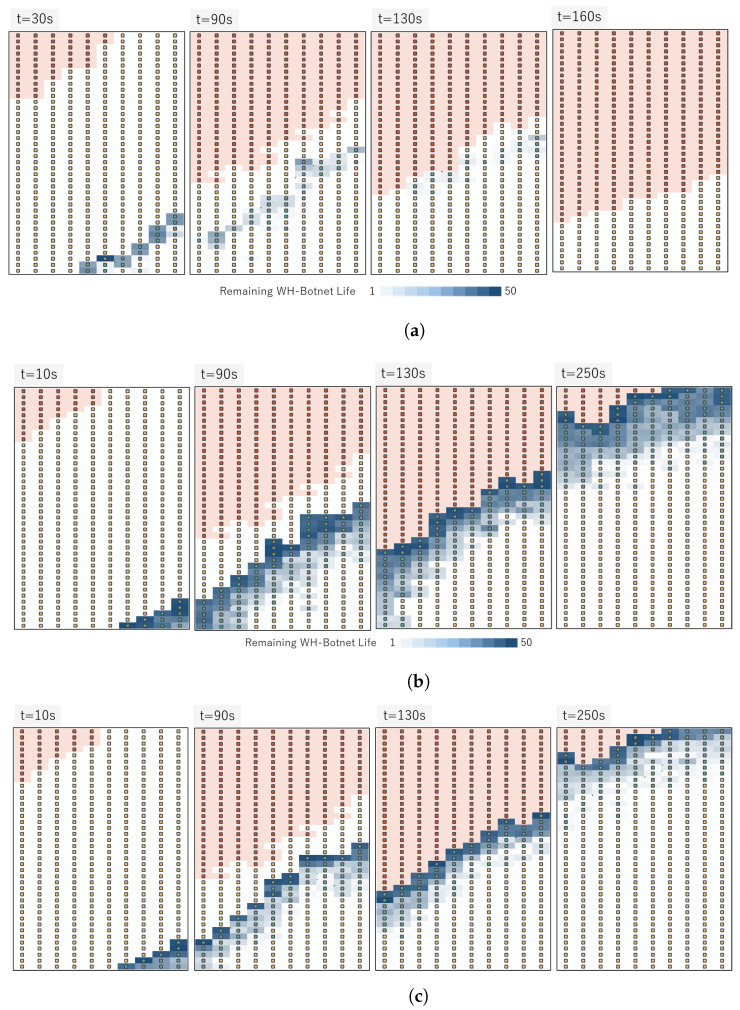
Comparison of diffusion range using different parameters. BDS launcher positioned at right-bottom. (**a**) Result of remaining botnet life $l_{n}$ (L=10,α=10, β=6). (**b**) Result of remaining botnet life $l_{n}$ (L=10,α=10, β=3). (**c**) Result of remaining botnet life $l_{n}$ (L=10,α=5, β=1).

**Figure 10 sensors-23-01018-f010:**
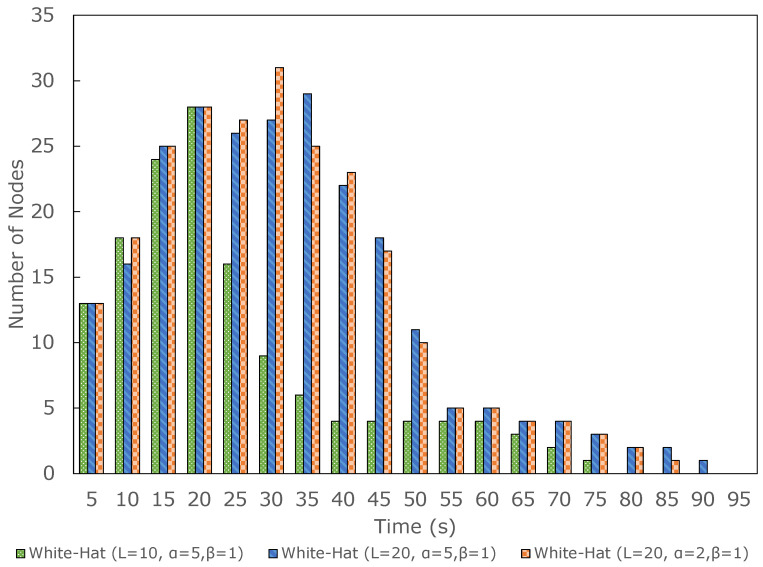
Comparison of the number of white-hat botnet’s successful parameter settings.

**Table 1 sensors-23-01018-t001:** Comparison of Related Works.

Literature	Method	Detection	Prevention	Mitigation	Clean-Up	Static Lifespan	Dynamic Lifespan
Namanya et al. [11]	Signature-based	*√*					
Kakisim et al. [12]	Signature-based	*√*					
Rehman et al. [13]	Signature-based	*√*					
Vasan et al. [14]	Machine Learning	*√*					
Liu et al. [15]	Machine Learning	*√*					
Botacin et al. [16]	Signature-based	*√*	*√*				
Hussain et al. [17]	Machine Learning	*√*	*√*				
Albanese et al. [18]	Moving Target Defense		*√*				
Amich et al. [19]	Moving Target Defense		*√*				
Hwang et al. [20]	Emulation		*√*				
Sajjad et al. [21]	Emulation		*√*				
Ajmal et al. [22]	Emulation		*√*				
Chu et al. [23]	Machine Learning	*√*		*√*			
Martinelli et al. [24]	Machine Learning	*√*		*√*			
Kepner et al. [25]	Forward Defense	*√*		*√*	*√*		
Yamaguchi et al. [26]	BDS Launcher	*√*		*√*	*√*	*√*	
Pan et al. [27]	BDS Launcher	*√*		*√*	*√*	*√*	
Our Work	BDS Launcher	*√*		*√*	*√*		*√*

**Table 2 sensors-23-01018-t002:** Comparison of parameter settings. Yes: Succesful Diffusion; No: Unsuccessful Diffusion.

Experiment Settings	α	β	L=5	L=10	L=20
**Fixed Initial Lifespan**	**N/A**	**N/A**	No	No	**Yes**
**Balanced Small Attack—Withdrawal**	**2**	**2**	No	No	**Yes**
**Small Attack—Small Withdrawal**	**2**	**1**	No	No	**Yes**
**Large Attack—Small Withdrawal**	**5**	**1**	No	**Yes**	**Yes**
**Large Attack—Medium Withdrawal**	**5**	**2**	No	**Yes**	**Yes**
**Balanced Large Attack—Withdrawal**	**5**	**5**	No	No	No

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
