# Peer review of "Diffusion of White-Hat Botnet Using Lifespan with Controllable Ripple Effect for Malware Removal in IoT Networks"

_sensors, 2023, doi:10.3390/s23021018_

Round 1
Reviewer 1 Report
The authors proposed a method to control the diffusion of white-hat botnets that can assist in removing malware for IoT Networks. The manuscript is well-structured, but the following facts should be addressed.
- In the abstract, the authors should add some facts, like 1) how could the white-hat botnets assist to remove the malware, and 2) why is it needed to think about the diffusion concern of the white-hat botnet.
- According to Table 1, your proposal can’t detect and prevent the malware. It should be clear how to mitigate and clean up them, without even doing these things.
- In the case of mitigation and cleaning up the malware, you just compared it with your work. It would be better if the related works of others are added to this manuscript.
- The more important thing is that almost of IoT nodes are resources constraint. If the white-hat botnet spread on these nodes, is it possible to handle the payload on resource constraint devices?
- In section 2, the explanation of Table 1 should be expressed as “Table 1”, rather than using the term, “Table Raf”.
- The more related works should be referenced to describe the white-hat botnets, their pros, and cons.
Reviewer 2 Report
Overall Comments:
This paper discusses the Diffusion of White-Hat Botnet Using Lifespan with Controllable Ripple Effect for Malware Removal in IoT Networks.
*Recommended changes:
· The abstract needs rewriting as the problem statement is not very clear. The sentence structure also needs improvement.
· You must always start the abstract with background and then jump on to the proposed methods.
· Introduction; started very good however it lacks in depth background and related discussion. Add more discussion after line 23.
· Line 27; although it's good you have provided references but discuss in a couple of lines about the ‘three-step’ method.
· Line 58; Table needs index reference [1]. Edit and update.
· Good use of references in the literature review. Make sure the table appears after you mention it for the first time in the text. Try to add more papers from the recent past.
· Although the Problem identification is good however section 3 seems more like a methods chapter. Update/merge as appropriate.
· Section 4; good use of figures and simulation diagrams. Make sure you reference these efficiently in the main text as appropriate.
· Figure 7; Good results. Try and change the graph patterns for the reader if they print in greyscale.
· Why the ‘L’ values are set to 5, 10 and 20 and not something else; explain it in the main text and refer it accordingly.
· Is this approach suitable for adversarial attack detection?
· The conclusion needs rewriting as the aims which were identified at the start of the paper are not necessarily understandable from the conclusion. It needs more elaboration.
· Authors should specify what plans they have for the future related to the outcomes of this paper with more detail.
· Academic writing style needs significant improvements.
